# Reactive Oxygen Species (ROS)-Mediated Antibacterial Oxidative Therapies: Available Methods to Generate ROS and a Novel Option Proposal

**DOI:** 10.3390/ijms25137182

**Published:** 2024-06-29

**Authors:** Silvana Alfei, Gian Carlo Schito, Anna Maria Schito, Guendalina Zuccari

**Affiliations:** 1Department of Pharmacy (DIFAR), University of Genoa, Viale Cembrano, 4, 16148 Genoa, Italy; guendalina.zuccari@unige.it; 2Department of Surgical Sciences and Integrated Diagnostics (DISC), University of Genoa, Viale Benedetto XV, 6, 16132 Genoa, Italy; giancarlo.schito@unige.it (G.C.S.); amschito@unige.it (A.M.S.)

**Keywords:** multidrug-resistant pathogens, antimicrobial oxidative therapy, reactive oxygen species (ROS), biochar (BC), biochar-derived permanent free radicals (PFRs), advanced oxidation processes (AOPs)

## Abstract

The increasing emergence of multidrug-resistant (MDR) pathogens causes difficult-to-treat infections with long-term hospitalizations and a high incidence of death, thus representing a global public health problem. To manage MDR bacteria bugs, new antimicrobial strategies are necessary, and their introduction in practice is a daily challenge for scientists in the field. An extensively studied approach to treating MDR infections consists of inducing high levels of reactive oxygen species (ROS) by several methods. Although further clinical investigations are mandatory on the possible toxic effects of ROS on mammalian cells, clinical evaluations are extremely promising, and their topical use to treat infected wounds and ulcers, also in presence of biofilm, is already clinically approved. Biochar (BC) is a carbonaceous material obtained by pyrolysis of different vegetable and animal biomass feedstocks at 200–1000 °C in the limited presence of O_2_. Recently, it has been demonstrated that BC’s capability of removing organic and inorganic xenobiotics is mainly due to the presence of persistent free radicals (PFRs), which can activate oxygen, H_2_O_2_, or persulfate in the presence or absence of transition metals by electron transfer, thus generating ROS, which in turn degrade pollutants by advanced oxidation processes (AOPs). In this context, the antibacterial effects of BC-containing PFRs have been demonstrated by some authors against *Escherichia coli* and *Staphylococcus aureus*, thus giving birth to our idea of the possible use of BC-derived PFRs as a novel method capable of inducing ROS generation for antimicrobial oxidative therapy. Here, the general aspects concerning ROS physiological and pathological production and regulation and the mechanism by which they could exert antimicrobial effects have been reviewed. The methods currently adopted to induce ROS production for antimicrobial oxidative therapy have been discussed. Finally, for the first time, BC-related PFRs have been proposed as a new source of ROS for antimicrobial therapy via AOPs.

## 1. Introduction

The incessant and rapid increase of multidrug-resistant (MDR) pathogens causes the emergence of difficult-to-treat infections with long-term hospitalizations, high costs, and a frightening incidence of death. In the United States, more than 2.8 million antibiotic-resistant infections occur each year, resulting in 35,000 deaths [1]. Resistant bacteria are becoming an uncontrollable worldwide hazard to both humans and animals [2,3]. Management of antimicrobial resistance through available antibiotics is a global public health problem and represents a daily challenge for experts in the field. To limit the global antibiotic resistance crisis, the main solution would be to reduce the volume of the unscrupulous use of antibiotics both in medicine, agriculture, and the environment [4], as well as to perform an efficient infection control strategy to prevent the spread of contagions. Anyway, the development of novel antimicrobial drugs remains urgent and mandatory [1]. Although several new agents based on existing classes of antibiotics are being developed, there has been little advancement in the exclusive discovery of novel agents [4].

Moreover, the capability of several bacterial and fungal species to form biofilms is an alarming mechanism through which pathogens develop a very complex form of resistance. Biofilm-producing pathogens represent a significant problem in many clinical settings since biofilm further increases their tolerance towards conventionally prescribed antimicrobials [5,6]. The high-dose use of antibiotics in biofilm conditions to treat chronic wounds, burns, chronic respiratory diseases, cystic fibrosis, and recurrent cystitis leads to intense selective pressure, which paradoxically drives further antibacterial resistance [4].

In this alarming scenario, the need for the development of optional per se effective therapeutic approaches or of alternative treatments that can improve the antimicrobial efficacy of existing drugs as well as biofilms is imperative [7]. This second strategy would reduce the amount of antibiotics to be used, thus limiting the emergence of resistance in pathogens. Entirely novel antimicrobial instruments characterized by a unique mechanism of action are represented by reactive oxygen species (ROS) induced by different methods [4].

### 1.1. Reactive Oxygen Species (ROS)

ROS have demonstrated in vitro and in vivo a significant antimicrobial action against a wide spectrum of Gram-positive and Gram-negative organisms, including MDR isolates and biofilm-producing pathogens [8]. The use of ROS could represent a new therapeutic approach for topical use on skin, mucosal membranes, or internal tissue that may be colonized with microbial inhabitants and biofilms [9].

Treatments involving ROS as antimicrobial agents are already available for topical application, are clinically approved to treat infected wounds, and are being developed for clinical use in other settings [10]. 

As mentioned above, ROS is the well-known acronym used in several sectors, including medicine, to indicate reactive oxygen species, including radical and not radical oxygen-containing atoms and molecules, such as superoxide anion (O_2_^−^), singlet oxygen (^1^O_2_), peroxide (O_2_^−2^), hydrogen peroxide (H_2_O_2_), hydroxyl radicals (OH•), and hydroxyl anions (OH^−^), that are constantly being formed as byproducts of the physiologic aerobic metabolism of cells [11]. Additionally, ROS can react with NO produced by cells from intracellular *L*-arginine via the action of epithelial nitrogen oxide synthetase (NOS), neuronal NOS, and inducible NOS, forming other reactive species such as NO• and ONOO^−^, while NO• in combination with O_2_ provides ONOO•. These molecules are referred to as reactive nitrogen species (RNS) [11].

#### 1.1.1. Oxidative Stress (OS) by ROS

In normal conditions, ROS and RNS generation is kept under control by the antioxidant defenses and repair systems of cells [12]. On the contrary, when overproduced, the detoxification systems of cells fail to maintain ROS and RNS physiological levels, which accumulate, thus causing the onset of oxidative stress (OS) and inflammation. Irreversible damage to DNA, lipids, and proteins occurs, thus promoting aging, age-related diseases, and several degenerative human disorders [13].

#### 1.1.2. Oxidative Stress (OS) by ROS Is the Cause of Diseases in Humans

Collectively, OS is a cascade of events that frequently triggers and accompanies molecular/cellular pathogenic events. It is responsible for several human disorders, including carcinogenesis [14,15], atherosclerosis, cardiovascular, and neurodegenerative diseases [16,17].

#### 1.1.3. Oxidative Stress (OS) by ROS Is the Cause of Diseases in Microorganisms

As occurs in humans and also in pathogens, OS builds up when prooxidants overpower antioxidants. Therefore, ROS get accumulated in the microorganism’s cell, thus exceeding the cell’s capacity to readily detoxify them [18].

As examples, the host immune response as well as several antimicrobials counteract infections by inducing ROS accumulation. While the interaction between the host and pathogens causes exogenous OS in bacteria, intracellular redox reactions, antibiotics, and uncontrolled aerobic respiration contribute to endogenous OS [19]. ROS cause multiple damages to the bacterial cells, including double-stranded breaks in DNA by oxidizing dCTP and dGTP pools, which results in the misincorporation of bases into DNA. Additionally, ROS induces lipid peroxidation and protein carbonatization [20], thus exerting a very rapid bactericidal activity. It has been reported that ROS were able to cause a 3 log CFU reduction in 30 min and total eradication in 2 h when used against *Staphylococcus aureus* [21]. Several conventional and alternative antibiotics, including metal nanoparticles and natural molecules, exert their antimicrobial properties by inducing ROS hyperaccumulation in pathogens [22]. Other methods to exert ROS-mediated antimicrobial effects include photodynamic therapy (PDT), honey reactive oxygen (HRO) therapy, and hyperbaric oxygen treatment (HBOT).

#### 1.1.4. ROS as a New Weapon against Pathogens

On these considerations, ROS could really represent an effective option for eradicating MDR pathogens. While procedures to induce ROS in microbial cells, such as HBOT and PDT, are traditional, the use of nanomaterials and engineered medical honey are rather novel and promising methods. Nevertheless, nanotoxicology is a field that is still not clearly defined and lacks sufficient and unequivocable epidemiologic data, information, and regulation. Furthermore, such methods may induce ROS formation in host cells as well. To make possible an enlargement of the clinical use of ROS to counteract MDR pathogens and related biofilm, the development of other delivery strategies for increasing the selectivity of ROS for microbial pathogens over the host tissue is necessary. In this regard, based on our recent studies on biochar (BC) [23,24], we profit from this review to propose a possible innovative method to be studied to produce ROS from a natural, low-cost source. Biochar (BC) is a carbonaceous material obtained by pyrolysis of different vegetable and animal biomass feedstocks and waste at 200–1000 °C in the limited presence or absence of oxygen. Due to its strong adsorption capacity, BC can exert a plethora of beneficial effects, including the removal of environmental pollutants and xenobiotics, thus preventing their uptake in plants, animals, and humans [25,26,27,28]. In microbiology, BC has been demonstrated to be helpful in limiting antimicrobial resistance by degrading/removing residual antibiotics from soil and water [24]. The so-called environmentally persistent free radicals (EPFRs) are known to exist in significant concentrations in atmospheric particulate matter (PM) and are primarily emitted from the combustion and thermal processing of organic materials. While their existence in combustion has been known for over half a century, only recently has their presence in environmental media and their healthy and/or hazardous effects been researched [29]. Nowadays, it has been demonstrated that BC can also contain persistent free radicals (PFRs) bound to the external or internal surfaces of its solid particles [29]. PFRs are reactive species due to unpaired electrons that can persist for several months, in contrast to traditional transient radicals [24]. Studies reported that PFRs are the main reason for BC’s capacity to degrade organic pollutants through the generation of active oxygen species (ROS) and sulfate radicals [23]. It was reported that the generated ROS, including radical (•OH, •O_2_^−^, •O_2_H, SO_4_^•−^) and non-radical species (^1^O_2_), successfully degraded several organic pollutants, hormones, and eDNA by advanced oxidation processes (AOPs). Interesting, PFRs-mediated ROS showed antibacterial effects against *Escherichia coli* and *S. aureus* [30,31,32], thus supporting the idea of the possible use of BC-derived PFRs as a novel method to induce ROS generation for antimicrobial oxidative therapy. In the following sections, all that was introduced in this section will be reviewed and discussed in greater depth.

## 2. Reactive Oxygen Species (ROS) and Oxidative Stress (OS)

### 2.1. Physiological and Pathological Origins of ROS

The following Figure 1 schematizes the main endogenous processes by which ROS can form in cells and the detrimental effects they can have on health [11], including DNA damage, lipids, and protein peroxidation, telomere reduction, aging, and death.

On the other hand, the following Table 1 collects the endogenous molecules, organelles, and metabolic processes responsible for ROS production, dividing them into enzymatic and non-enzymatic ones. Also, it reports the sources, external to cells, that can induce ROS formation, and it refers to the main radical and non-radical oxygen and nitrogen reactive species that can form upon these events.

Based on the recently acquired knowledge and literature reports [33,34,35,36], with respect to the Table reported by us in 2020 [11], environmental persistent free radicals (EPFRs) and biochar-related persistent free radicals (BC-PFRs) have been included among the exogenous sources of ROS in Table 1. Upon their formation, according to reported processes and mechanisms, they can induce ROS formation by reacting with atmospheric or water dissolved O_2_, as well as with H_2_O_2_ and/or persulfate [24]. As shown in Figure 1, the molecular oxygen from different sources can be reduced to the radical superoxide anion (O_2_^•−^), which is considered the primary ROS. Then, it reacts with other molecules through enzymatic or non-enzymatic metal-catalyzed processes, thus generating secondary ROS. In particular, phagocytic cells (neutrophils, monocytes, or macrophages) use NOX (Table 1) for one-electron reduction of molecular oxygen to the radical superoxide anion (O_2_^•−^) during cellular respiration.

Then, O_2_^•−^ is mainly transformed by superoxide dismutase (SOD) into hydrogen peroxide (H_2_O_2_), from which the highly reactive ROS hydroxyl ion (•OH) and radical HOO• are formed through the Fenton or Haber–Weiss reactions in the presence of transition metals (Figure 1). O_2_^•−^ is also produced from the irradiation of molecular oxygen with UV rays, photolysis of water, and by exposure of O_2_ to organic radicals formed in aerobic cells such as NAD•, FpH•, semiquinone radicals, cation radical pyridinium, or hemoproteins (Table 1).

While the radical O_2_^•−^ does not react directly with lipids, polypeptides, sugars, or nucleic acids, •OH and HOO• react especially with phospholipids in cell membranes and proteins, thus causing oxidative damage, DNA damage, telomere reduction, aging, and apoptosis [11] (Figure 1).

Furthermore, H_2_O_2_ can be converted by MPO (Table 1) to hypochlorous acid, which is particularly hazardous for cellular proteins [37].

Additionally, ROS can react with NO produced from intracellular *L*-arginine by cells as a defense mechanism, using three different kinds of NOS, such as epithelial NOS, neuronal NOS, and inducible NOS, thus forming reactive nitrogen species (RNS) such as NO• and ONOO^−^. Finally, NO• in combination with O_2_, can provide ONOO•, which induces lipid peroxidation in lipoproteins [11,38,39,40] (Table 1).

Some of the most representative oxygen and nitrogen reactive species reported in Table 1 have been correlated with their specific sources and with their physiological function in biological aerobic systems in Table 2.

Anyway, whatever their origin, both ROS and RNS cause indifferently detrimental oxidative modifications of cellular macromolecules such as carbohydrates, lipids, proteins, DNA, and RNA. Upon this damage, particular molecules are produced, which are considered markers of OS. The following Table 3 summarizes the main molecular targets of ROS and RNS, the reactions occurring during the damaging process, and the compounds that are consequently produced considered biomarkers of OS.

To counteract the detrimental effects reported in Table 3, cells have developed several repair systems able to restore or eliminate lipids, proteins, and DNA damaged by the action of ROS and RNS. Particularly, cytosolic and mitochondrial enzymes, which include polymerases, glycosylases, and nucleases, repair the damaged DNA, while proteinases, proteases, and peptidases, which are part of the proteolytic enzymes, remove damaged proteins. In addition, biological systems have developed both physiological and biochemical mechanisms to limit free radicals’ production and reactive species toxicity. At the physiological level, the microvascular system exerts the function of maintaining the levels of O_2_ in the tissues, while at the biochemical level, a protective activity is exerted both by endogenous (enzymatic and non-enzymatic) and exogenous molecules, as reported in Table 4 and Table 5. Collectively, GSH-Px, GR, and MSR are the main intermediaries in the processes for repairing oxidative damage.

### 2.2. Pathogen Responses to OS

As reported previously in the introduction, ROS cause multiple damages to the bacterial cells [22]. Anyway, bacteria can evade OS by several means, including detoxifying methods using enzymes such as catalase, alkyl hydroperoxide reductase, thioredoxin, and superoxide dismutase (SOD). Additionally, they use pigments such as carotenoids, metal homeostasis, and repair devices including DNA restoration, general stress response, and SOS response [18,46]. All these mechanisms are regulated by gene networks [46]. *E. coli* reacts with OS, mainly producing SOD and catalase, which convert O_2_^•−^ to H_2_O_2_ (SOD) and, in turn, H_2_O_2_ into H_2_O and O_2_ (catalase). While mammalian cells possess two types of SOD, *E. coli* owns three isoforms of SOD characterized by different metal *cores*. Particularly, sodA contains Mn, sodB includes Fe, and sodC comprises both Cu and Zn. *E. coli* also has two types of catalases, namely hydro-peroxidase I and hydro-peroxidase II [47]. Also, *E. coli* has several major regulators activated during OS, such as OxyR, SoxRS, OhrR, and RpoS. OxyR and SoxR control the catalase and SOD transcription in relation to the O_2_^•−^ and H_2_O_2_ concentrations. They undergo conformation changes when oxidized in the presence of hydrogen peroxide and superoxide radicals, respectively, and subsequently control the expression of cognate genes [48]. In contrast, the RpoS regulon is induced by an increase in RpoS levels. It is a specialized sigma factor that govern the expression of genes that lead to general stress resistance in cells [49]. These genes may be involved in eliminating oxidative agents, repairing systems of affected biomolecules, and maintaining normal cellular physiologic circumstances. Despite the enormous genomic diversity of bacteria, OS response regulators present in *E. coli* are functionally conserved in a wide range of bacterial groups. Bacteria have developed complex, adapted gene regulatory responses to OS, probably due to the high level of ROS produced endogenously through their basic metabolism. Additionally, several bacterial pathogens prevent the increase of ROS by directly inhibiting the synthesis of NADPH oxidase [50]. Iron homeostasis and remodeling of metabolism are two other methods by which bacteria lessen the damage caused by ROS. Bacteria can remodel their metabolism via upregulation of the glycoxylate shunt, thus reducing endogenous ROS formation, or by redirecting the metabolism toward the pentose phosphate pathway and augmenting the production of NADH, which refills the level of antioxidants. Ketoacids such as pyruvate and α-ketoglutarate can decarboxylate in the presence of ROS, thus originating toxic molecules and diminishing damage caused by ROS [51]. Iron, which is also involved in ROS generation by Fenton reactions, is crucial for the growth and survival of bacteria, and paradoxically, iron acquisition by siderophore action is pivotal to counteracting OS [52]. Siderophores are compounds that bacteria produce when intracellular iron concentrations are low to facilitate their uptake [53]. Two siderophores, namely Staphyloferrin A and B, have been found to enhance the resistance to OS in *S. aureus*, while *E. coli* produces an enterobactin siderophore to alleviate damage from OS [51,52]. It was demonstrated that OS in turn regulates bacterial siderophore production [53]. When *E. coli* was exposed to H_2_O_2_ and paraquat, the expression of enterobactin increased in the presence of a high concentration of iron, which reduced the sensitivity of the isolate to both H_2_O_2_ and paraquat [53]. Similarly, when methicillin-resistant *S. aureus* (MRSA) was exposed to the antimicrobial surface coating AGXX^®^, siderophore biosynthesis genes were highly upregulated. Some bacterial species produce biofilm as a highly specialized and organized form of resistance in which bacteria cooperate and stay protected by a self-produced biomass [54]. Persisters are bacterial cells in a dormant state with low metabolic activity existing in biofilm that showcase high antibiotic tolerance, can recolonize post-therapy [55], are less sensitive to ROS, and have demonstrated increased expression of efflux pumps. Efflux pumps major expression is another mechanism to react to OS, which allows bacteria to pump out the ROS-damaged proteins [56].

## 3. Antimicrobial Oxidative Therapies: Available Methods to Induce ROS Formation

As previously reported in the Introduction, the induction of high levels of reactive oxygen species (ROS) by several procedures, thus causing OS detriment to bacterial cells, has been extensively studied to inhibit several species of Gram-positive and Gram-negative bacteria, viruses, and fungi. The use of ROS represents a new therapeutic approach for topical use on skin, mucosal membranes, or internal tissue that may be colonized with microbial inhabitants and biofilms [57]. It was found that the antibacterial effect of several conventional and alternative antibiotics, metal nanoparticles, and natural molecules is also based on their capability of inducing ROS hyperaccumulation in pathogens [22]. Therefore, other methods have been developed, are clinically applied, or are in clinical trials to produce ROS finalized for antimicrobial oxidative therapy. They include photodynamic therapy (PDT), honey reactive oxygen (HRO) therapy, and hyperbaric oxygen treatments (HBOT) [57].

### 3.1. ROS Formation Induced by Conventional, Alternative, and Natural Antimicrobials

Before reviewing the main antimicrobial therapies based on ROS induction, such as PDT, HRO, and HBOT, in this section we have reviewed other methods to provoke ROS improvement for antimicrobial uses. Table 6 reports information concerning conventional and alternative antimicrobials, including some antibiotics, nanoparticles, and natural compounds, which exert their effects by generating ROS.

#### 3.1.1. ROS Formation Induced by Antibiotics

Clinically approved antibiotics such as erythromycin, by protein synthesis inhibition, and rifampicin, by inhibiting RNA synthesis, are effective against *Rhodococcus equi*, while vancomycin is antibacterial against *R. equi*, *Mycobacterium tuberculosis*, and *S. aureus*, by inhibiting cell wall synthesis inhibition. Moreover, norfloxacin, by inhibiting DNA gyrase, is effective against *R. equi*, *S. aureus*, and *E. coli*. Clofazimine, by DNA replication inhibition, and ethambutol and isoniazid, by cell wall synthesis inhibition, are antibacterial against *M. tuberculosis.* Finally, quinones, by different cellular targets, are active on *Enterococcus* sp., *Streptococcus* sp., *Staphylococcus* sp., and *Moraxela catarrhalis* [64]. Anyway, studies observed that antibiotics functioning with a primary mode of action not correlated with OS, interfering with some bacterial cell targets, as above reported, were found to cause bacterial damage while also generating ROS [71,72]. As shown in Figure 2, the interaction of antibiotics with bacterial cell targets can cause both ROS hyperproduction and cell damage. The damage and disease induced by the initial ROS hyperproduction cause, in turn, additional ROS induction and production. The self-sustained growth of ROS concentration in bacterial cells goes out of control, thus causing irreversible OS and lethally amplifying cellular damage, leading to bacteria death.

Particularly, some antibiotics generate ROS through overstimulation of electrons via the tricarboxylic acid cycle and the release of iron from the iron-sulfur clusters, thus activating the Fenton chemistry. Nitrofurantoin and Polymyxin B are two commonly used ROS-mediated antibiotics [22]. Nitrofurantoin, used to treat urinary tract infections by *E. coli* [18], acts through a NADH-dependent reduction, producing nitroaromatic anion radicals. The autooxidation of these anion radicals in the presence of O_2_ produces O_2_^−^, which ROS generate, thus causing OS and toxicity in bacteria [18]. Polymyxin B (PMB) is part of the family of antimicrobial peptides and is active mainly on Gram-negative bacteria such as *A. baumannii*, *P. aeruginosa*, and carbapenemase-producing *Enterobacteriaceae* [18,73]. Due to its neurotoxic and nephrotoxic properties, PMB is advised to be used only as a last resort antibiotic [18]. Sampson et al. demonstrated that PMB, in addition to being a membrane disruptor [74], induced cell death in Gram-negative bacteria by the accumulation of OH• [58]. Anyway, Arriaga-Alba and co-workers were the first authors to report oxidative stress induction as a part of the mechanism of action of nalidixic acid and norfloxacin in *Salmonella typhimurium*. [59]. Antibiotics were shown to upregulate many oxidative stress genes in *P. aeruginosa* [22]. Wang and Zhao found out that norfloxacin was more lethal in *E. coli* deficient in the catalase gene katG than in its isogenic mutants, thus confirming a potential pathway linking hydroxyl radicals to antibiotic lethality [60]. Also, ampicillin and kanamycin showed increased lethality in an alkyl hydroperoxide reductase ahpC *E. coli* mutant, lacking the defense system to contrast hydroxyl radicals. These studies evidenced increased superoxide levels in the bacterium, which were the source of H_2_O_2_, which in turn generated the highly toxic hydroxyl radical responsible for the improved lethality of antibiotics [60]. Hong et al. demonstrated that *E. coli* exposed to lethal oxidative stressors caused by antibiotics including nalidixic acid, trimethoprim, ampicillin, and aminoglycosides did not die only during the actual treatment but also post-treatment, after the removal of the initial stressor, due to post-stress ROS-mediated toxicity [61,62]. Anyway, the connection between antibiotic action and ROS is not yet clearly demonstrated. Since it was demonstrated that antibiotics also work under anoxic conditions, some reports oppose the idea that the generation of ROS contributes to their lethality, which is instead influenced by the bacterial metabolism, iron homeostasis, and iron-sulfur proteins [75,76,77]. Although contradictory studies exist concerning the possible ROS influence on antibiotic lethality, it has been established that numerous alternative antimicrobials work by inducing ROS-mediated OS in bacteria. Unfortunately, some ROS-producing antibiotics, such as aminoglycosides, fluoroquinolones, and β-lactam antibiotics, may induce host cellular damage in specific tissues, such as the renal cortex or tendons, by generating OS, which is anyway manageable by specific antioxidant molecules [64].

#### 3.1.2. ROS Formation Induced by Alternative Antimicrobials

Several novel alternative antimicrobials are under development whose primary mode of action seems to be through the generation of ROS-causing OS in bacteria. Generally, these antimicrobials often target the redox defenses, such as the thiol-dependent enzyme thioredoxin reductase (TrxR) in bacteria [63]. Examples of ROS-mediated antimicrobials include Ebselen, nanoparticles, nanozymes, and AGXX^®^. Even if not reported in Table 6, due to their photodependent capability to produce ROS, emerging nanomaterials such as carbon dots (CDs), produced by different sources, have demonstrated antimicrobial properties by ROS induction.

Ebselen is an organo-selenium-based antioxidant drug endowed with anti-inflammatory, antioxidant, and cryoprotective effects. It acts by inhibiting TrxR in bacteria lacking glutathione, thus triggering OS thus being lethal to these pathogens. Ebselen was recently shown to efficiently inhibit in vitro the growth of MDR *S. aureus*, to improve wound healing in rats, and to reduce the bacterial load in *S. aureus* skin lesions in rats [63]. Ebselen has been reported to inhibit *M. tuberculosis*. Importantly, Ebselen could also be combined with other ROS-stimulating compounds that block the antioxidant defenses of bacteria, such as silver nanoparticles (NPs) [64].

Mainly due to their small size (<100 nm), nanoparticles (NPs) can cause hyperproduction of ROS, which can cause carbonylation of proteins, peroxidation of lipids, DNA/RNA breakage, and membrane structure destruction, thus damaging cells [78]. Among NPs, those made of silver, such as silver oxide NPs (AgNPs), titanium dioxide, silicon, copper oxide, zinc oxide, gold, calcium oxide, and magnesium oxide NPs, have been ported to have antibacterial effects against both Gram-positive and Gram-negative pathogens [57]. Mesoporous silica NPs (MSNPs) containing a maleamato ligand (MSNPs-maleamic) and others containing also copper (III) coordinator ions (MSNPs-maleamic-Cu), synthesized by Diaz-Garcia et al., demonstrated antibacterial activity against *E. coli* and *S. aureus* by OS induction [65]. The minimum inhibitory concentration values (MICs) of MSNPs-maleamic and MSNPs-maleamic-Cu established that both preparations performed better against *E. coli* than on *S. aureus* and that MSNPs-maleamic (MIC = 62.5 µg/mL) was more effective than MSNPs-maleamic-Cu (MIC = 125 µg/mL) against *E. coli* [65]. Both preparations caused a significant increase of ROS in both species (30–50% more than in control), and the NPs that caused the major increase displayed lower MICs (MSNPs-maleamic, 50% in *S. aureus*, and 40% in *E. coli*), thus confirming that ROS and OS generation contribute to the antibacterial mechanism of action of MSN-maleamic and MSN-maleamic-Cu [65]. As recent members of the nanomaterial family, carbon dots (CDs) have demonstrated photoluminescence, easy surface functionalization modification, simple preparation, low toxicity, low side effects, and a lower probability of developing resistance, showing great antibacterial and antiviral potential [79]. Although the specific antibacterial mechanism of CDs needs to be strengthened, several studies have associated their antimicrobial effects with ROS improvements. Rabe et al. prepared positively charged CDs with different surface passivation layer thicknesses using polyethylene imine (PEI) of different molecular weights, which demonstrated strong antibacterial activity by photogenerated ROS [80]. Bing et al. prepared both positively charged SC-CDs (+27.6 mV), negatively charged CC-CDs (−19.5 mV), and neutrally charged GC-CDs (0.946 mV) [81]. When tested on *E. coli*, these CDs demonstrated antibacterial effects based on ROS aggregation, cell apoptosis, and bacterial cell membrane destruction, which led to programmed bacterial death [81]. Moreover, when CDs with a high negative surface charge (−75 ± 4 mV) were used to treat *S. aureus* and MRSA, they showed antibacterial activity under laser irradiation. Particularly, upon CD adhesion to the cell surface, production of ROS and cell wall damage occurred, protein structure and function changed, with the subsequent death of bacteria [82,83]. Finally, but many other examples exist, Wang et al. synthesized graphene-based Cl-doped CDs, which, due to the high content of defect sites caused by Cl doping, were capable of producing ROS under visible light irradiation and were exploitable for antibacterial applications [84].

The major drawback of ROS-based antibacterial therapy is the low selectivity of ROS, which is detrimental to human cells as well. Recently, nanomaterials possessing enzyme-like characteristics and referred to as nanozymes (NZs), have been reported to produce surface-bound ROS that were selective in killing bacterial cells over mammalian ones [66]. Particularly, ROS bound on silver and palladium bimetallic alloys (AgPd0.38) efficiently killed antibiotic-resistant bacteria, including *S. aureus*, *Bacillus subtilis*, *E. coli*, and *P. aeruginosa* (MBC = 4–16 µg/mL), without developing drug resistance and inhibiting biofilm formation [66].

AGXX^®^ is a silver and ruthenium-based antimicrobial surface coating that can be coated or deposited on various carriers, such as cellulose, plastics, ceramics, or metals. AGXX^®^ demonstrated low levels of toxicity to mammalian cells [85] and no tendency to develop resistance due to its multiple modes of action [67]. ROS are produced catalytically by AGXX^®^, which has been demonstrated to inhibit the growth of *Enterococcus faecalis* and MRSA, as well as to prevent biofilm formation in MRSA [54,68]. In both species, AGXX^®^ affected oxidative stress defenses such as superoxide dismutase (SodA), catalase (KatA), alkyl hydroperoxide reductase (AhpCF), thioredoxin/thioredoxin reductase (Trx/TrxR), disuphide reductase (MerA), and oxidized bacillithiol (BSH) [69]. AGXX^®^-induced OS, general stress, heat shock (expression of Clp proteases), and copper stress [54,68]. In MRSA, it affected iron homeostasis and upregulated several siderophore biosynthesis (sbn) genes [54,69], while in *S. aureus*, increased protein-thiol oxidations, protein aggregations, and a BSH redox state were observed [67,69]. The mechanism of action of AGXX^®^ was assessed in *S. aureus* by observing two interconnected redox cycles by which AGXX^®^ simultaneously exerts ROS-mediated (superoxide anion, hydrogen peroxide, and highly toxic HO•) antimicrobial effect and achieves self-renewal.

#### 3.1.3. ROS Formation Induced by Natural Compounds

In addition to synthetic compounds, natural compounds can exert ROS-mediated antimicrobial effects. It is the case of allicin and honey, with the latter being the main ingredient of a clinically approved gel formulation for topical administration. Honey is used in honey antimicrobial therapy, which has been discussed here in a dedicated section. There are many other secondary metabolites produced by plants that may elicit oxidative stress in bacteria, such as catechins, ferulic acid, and their derivatives [64]. The combination of other ROS- and RNS-generating antimicrobials with these compounds may lead to the development of promising therapeutic strategies against different intracellular bacterial pathogens [64]. Particularly, allicin is a constituent of garlic, which works as a thiol-reactive compound, decreasing the levels of low molecular-weight thiols, which should function as a defense against ROS. Their increase causes OS in bacterial cells, thus inhibiting their growth [70].

### 3.2. ROS Formation Induced by Antibacterial Photodynamic Therapy

In a study on the toxicity of small concentrations of acridine red on *Paramecium* spp., it was recognized that the observed toxicity was dependent on the time of day and the amount of daylight [86]. Later, von Tappeiner and Albert Jesionek clinically applied this approach to treat skin carcinomas and coined the term “photodynamic phenomenon” [87,88,89]. Thus, anticancer photodynamic therapy was born. In those years, the successful photodynamic inactivation of bacteria was also described [86]. Anyway, while anticancer PDT has been clinically applied for 25 years, at least in the treatment of actinic keratosis or basal cell carcinoma, its application as an antimicrobial option has only more recently been rediscovered to manage the emergence of the first drug-resistant infections in the healthcare sector during the early 1990s [90,91].

#### 3.2.1. Basic Principles of Photodynamic Therapy

PDT is based on the combination of three elements, including a non-toxic compound referred to as a photosensitizer (PS), light in a spectral range appropriate for exciting the PS (typically from the visible to near infrared (NIR) spectrum), and molecular oxygen [91]. The mechanism of PDT is described by the Jablonski diagram reproduced in Figure 3.

Briefly, upon absorption of a photon (A), the PS moves from its ground singlet state (S_0_) to an excited singlet state (Sn). In this status, PS can lose energy, thus returning to S_0_ by emitting fluorescence (F) or heat (H) via internal conversion. On the contrary, it can pass to a longer-living excited triplet state PS (T_1_) through an inter-system crossing (ISC) process. Following, it can either return to the S_0_ state by phosphorescence emission (P) or by generating reactive oxygen species (ROS) by two mechanisms [92]. When a type I mechanism is followed, electrons are transferred to surrounding substrates, thus forming superoxide radical anions (O^2−•^), that undergo dismutation into hydrogen peroxide (H_2_O_2_), from which the highly reactive hydroxyl radical (HO•) derives via Fenton-like reactions. Differently, in the type II mechanism, energy and not charge are transferred directly to the ground-state molecular oxygen (^3^O_2_), leading to the emergence of singlet oxygen (^1^O_2_), which is nothing else than energized molecular oxygen [93]. At this point, PS is returned to its S_0_ status, ready to begin another cycle with the production of additional ROS. Interesting, one PS molecule can generate thousands of molecules of ^1^O_2_ before being destroyed. The singlet oxygen quantum yield describes the amount of type II mechanism [93]. During antibacterial PDT, type I and type II reactions can occur simultaneously, and the ratio between these processes depends on the type of PS utilized, its chemical structure, and the specific microenvironment in which the PDT is implemented. The interplay between type I and type II mechanisms is a critical factor to consider for an optimized treatment and understanding its underlying photochemical processes [94,95,96]. As already reported, ROS are detrimental for bacteria by targeting several vital microbial molecules, such as proteins, lipids, and nucleic acids, thus determining bacterial death. An additional type III reaction has been recently incorporated into the familiar categories of type I and II mechanisms. In this novel process, free radicals of inorganic compounds are generated, regardless of the presence of oxygen, which would participate in the photoinactivation of microorganisms [97].

#### 3.2.2. Photosensitizers Used in Clinical Trials

The main photosensitizers used in clinical trials are listed in Table 7. They include phenothiazinium, porphyrin, chlorin, phthalocyanine, xanthene derivatives, fullerenes, phenalenones, riboflavin, curcumin, hypericin, and 5-amino-lavulinic acid [98,99]. In this context, porphyrins are aromatic macrocycles that exhibit a characteristic absorption spectrum with a strong π–π* transition of ≈400 nm (Soret band) and four Q bands in the visible region. They are endowed with a strong ^1^O_2_ generation efficiency and an excellent fluorescence property. Particularly, Photofrin is recorded as the first-generation PS for PDT. Unfortunately, Photofrin suffers from poor water solubility and a low extinction coefficient in the NIR region [100]. Phthalocyanines (PCs) represent the second-generation of PSs for PDT. Compared with porphyrins, PCs have an exact molecular structure and better photophysical and photochemical properties. PCs exhibit a strong absorption band in the red region, and the presence of metal atoms, such as Zn, Al, and Si, yields a long T1 lifetime and a high ^1^O_2_ generation quantum yield [101]. Unfortunately, drawbacks such as a strong tendency to aggregate in aqueous solutions and a too slow clearance in vivo should be solved before their massive application in clinical PDT [100]. Organic dyes, such as indocyanine green (ICG), IR-825, and IR-780, showed considerable application in fluorescence imaging and PDT due to their near-infrared (NIR) absorption and excellent biocompatibility. Curcumin is a photoactive, polyphenolic compound derived from the turmeric root. Curcumin shows excellent phototoxicity to cancer cells and cytoprotectivity to normal cells. However, its poor water solubility and rapid clearance from the living body prevents its use in vivo. In this regard, the introduction of electron-donating groups to the curcumin skeleton can redshift the fluorescence wavelength, while modification with glycosylated ligands can significantly enhance its water solubility [102,103]. Fullerenes have peculiar electronic properties and biological activities. Specifically, C60 is an extremely efficient ^1^O_2_ generator with a quantum yield close to 100% [104]. However, its PDT application is limited because of its hydrophobic surface and extremely poor water solubility. Therefore, the development of novel methods to improve the water dispersibility of C60 has received considerable attention for the past few years [100].

The most common PSs clinically used in anticancer PDT have also been extensively studied for their antibacterial properties and effectiveness in the treatment of various bacterial infections. Many published studies have determined that phenothiazinium PSs such as MB and TB are effective on planktonic bacteria. Furthermore, some studies also tested the efficacy of phenothiazinium against biofilm structures [98]. Recently, new derivatives such as dimethyl methylene blue (derived from MB) and EtNBS (N-ethylpropylsulfonamido) have been studied. These dyes possess a high cationic charge, which makes them more effective against bacterial cells [98]. Rose Bengal is an anionic synthetic xanthene dye that has shown promise in several clinical trials for PDT, particularly in the treatment of localized bacterial infections. Other synthetic anionic xanthene dyes derived from Fluorescein are Eosin Y and Erythrosine (ERY). All these dyes have an absorption peak in the green wavelength range (480–550 nm). The attachment and uptake of anionic PSs by the bacterial cells are lower than cationic ones [98]. Amino-levulinic acid (ALA) is a prodrug that can be converted to protoporphyrin IX (PpIX), clinically used (in the form of hydrochloride salt) in combination with blue light illumination for the treatment of minimally to moderately thick actinic keratosis of the face or scalp. It has been demonstrated to be effective in the treatment of bacterial infections, including periodontal infections. Additionally, a variety of evidence has proven that 5-aminolevulinic acid-based photodynamic therapy (ALA-PDT) is clinically effective in the management of *Acne vulgaris* and is recommended as an alternative treatment modality for severe acne [130]. Concerning chlorins, mainly cationic derivatives of chlorin-e6 are used for PDT. Photodithazine^®^ is a commercially available chlorin-e6 derivative with two positive charges. Chlorine e6 (Ce6) and various phthalocyanines have strong antibacterial properties and have been used in preclinical studies and in some early clinical trials for antibacterial PDT. Curcumin is a natural compound found in *Curcuma longa*, and its cationic derivatives have also been investigated as possible PSs. Collectively, the use of specific PSs in clinical trials strongly varies depending on the target bacterial infection and the research objectives of each study [98,131,132].

#### 3.2.3. Antibacterial PDT vs. Antibiotics

Unfortunately, as reported in the following Table 8, even if the advantages of using antibacterial PDT (APDT) compared to antibiotics are significant, APDT has certain limitations that need careful consideration.

Despite promising results that have been observed in some diseases, the clinical translation of PDT for bacterial infections has progressed slower than for cancer treatment and leisurelier than anticipated. Therefore, its widespread adoption in medical practice remains limited. Among the limitations, the modest capability of light to penetrate skin, tissues, and organs hampers the application of APDT for systemic infection and reduces its effectiveness in topical treatments. The penetration of light depends on the optical properties of the tissue and the wavelength of the light used. There is heterogeneity between tissues and even within a tissue. These inhomogeneity sites (e.g., nuclei, membranes, etc.) cause light scattering, reflecting, transmitting, or absorption [133]. Light within the 620–850 nm spectrum range achieves optimum tissue penetration and PDT applications [133]. Mainly concerning potential side effects, understanding and effectively managing them is critical for optimizing patient outcomes. To fully harness the potential of antibacterial PDT as a valuable therapeutic strategy for combating bacterial infections, it is imperative to address these drawbacks through ongoing research efforts and comprehensive approaches [131,140]. The successful outcome of antibacterial PDT depends on the selected PS. To enhance PS selectivity over host tissue while maintaining nonselective activity against microbial species represents a pivotal challenge. The optimal PS should target both Gram-positive and Gram-negative species, accounting for their differential treatment responses. Several very recent studies have described the latest advancements in the field over the past years, emphasizing new PS systems relevant for antibacterial PDT’s methodologies [86,141,142,143], which have recently been reviewed by Sébastien Clément and Jean-Yves Winum [134].

#### 3.2.4. Light Sources

Different light sources have been and are employed in PDT, each with advantages and disadvantages, as reported in Table 9. In addition to the light sources reported in Table 9, “non-coherent” or “non-thermal” light sources without giving details of the actual light source used have been reported.

Additionally, there are also occasional reports of other light sources such as endoscopy systems, photopolymerisers, and supra-luminous diodes (SLD) [144]. All in all, for the irradiation of a given PS, parameters such as radiant exposure, light irradiance, power output, spectral emission, intensity of the respective light source, as well as the mode of light delivery (via optical fiber or directly), are more important than the type of the light source itself [144].

#### 3.2.5. Clinical Trials

Due to the multi-faceted nature of the antimicrobial photodynamic process and its diverse targets, the emergence of resistance in microbes becomes highly improbable. Consequently, the anti-bacterial PDT should be considered promising as a potent and innovative approach to combating bacterial infections [147]. In the last few years, the volume of research focused on PDT as a therapeutic device to inhibit a wide range of microorganisms, including bacteria, has remarkably increased. Anyway, the antibacterial PDT has yet to attain the capability to tackle systemic infections. On the contrary, it has demonstrated high potential for addressing localized infections sustained by MDR bacteria, including hospital-acquired pathogens such as *E. faecium*, *S. aureus*, *K. pneumoniae*, *A. baumannii*, *P. aeruginosa*, and *Enterobacter* spp., which together constitute the group ESKAPE [148]. Additionally, in vitro and in vivo studies have demonstrated the capability of PDT to eradicate or significantly reduce biofilms, thus finding applications in dental diseases, skin infections, and orthopedic implant treatment [149,150]. The ongoing advancement of antibacterial PDT systems is also marked by the continuous refinement of the strategies, particularly through synergistic combinations of diverse chemicals. Antibacterial PDT has been applied in about 40 clinical trials for the treatment of dermatological disorders and oral infections. Clinical trials on antibacterial PDT have focused on evaluating its safety and efficacy in treating specific bacterial infections, especially those that are challenging to manage with traditional antibiotics [91]. Some of the key findings from these trials include those obtained in dermatological disorders such as acne vulgaris and bacterial skin conditions, where antibacterial PDT studies have shown positive outcomes in reducing the severity of acne lesions [151,152,153,154]. Some studies have also explored applications for treating infected wounds, especially those associated with MDR bacteria. The results suggest that antibacterial PDT can be beneficial in promoting wound healing and controlling bacterial growth [155,156]. In the field of periodontal diseases, clinical trials have shown promising results for treating chronic periodontitis, a common bacterial infection that affects the gums and supporting structures of the teeth [157,158,159,160]. Other clinical trials have also examined the use of antibacterial PDT in managing bacterial infections related to medical implants, such as catheters and prosthetic devices [161,162].

### 3.3. ROS Formation Induced by Antibacterial Honey Therapy

Honey was used in traditional medicine mainly to treat wounds due to its antimicrobial effects and healing properties. The bactericidal efficacy of honey was reported more than a century ago by Van Ketel [163], whose findings prompted extensive research on honey over the next decades. With the advent of modern medicine, interest in honey and its medical use decreased [163]. Nowadays, honey is undergoing a revival in its consideration for antimicrobial and wound healing applications, due to the rising global antibiotic resistance, which makes the development of novel alternative therapies to combat infections necessary. Several factors can contribute to its effective antimicrobial activity, which can strongly vary depending on different microbial strains, including the geographical and botanical source, its harvesting, processing, and storage conditions. It has been demonstrated that a range of both Gram-positive and Gram-negative bacteria, including MDR strains, biofilms, fungi, and viruses, can be inhibited by honey. Furthermore, susceptibility to antibiotics can be restored when used synergistically with honey. Table 10 reports the antibacterial effects of honeys from different geographical sources and the related target bacteria.

#### 3.3.1. Medical Application of Honey

Although the knowledge of the antibacterial compounds involved in the antibacterial effects of honey remains incomplete, the information on honey has remarkably expanded in recent years. Otherwise, despite the variability of the antibacterial activity of honey, which limits its extensive applicability in medicine, several honeys have been approved for clinical application [179]. Currently, honey is mainly used as a topical antibacterial agent in wound applications. Its high viscosity grants an effective, hydrated, and protective barrier between the wound site and the external environment. Honey has been used to treat wounds such as burns, trauma, and chronic wounds, where the complex wound healing process could be interrupted by infection or specific disease states (e.g., diabetes), thus limiting the development of irreversible chronic wounds, recurrent infections, amputation/limb salvage, and life-threatening conditions [163]. For mild to moderate superficial and partial-thickness burns, honey was more effective than conventional treatment for reducing microbial colonization and improving wound healing [180,181]. In a study, the application of honey to tunneled cuffed hemodialysis catheters resulted in a comparable bacteremia-free period compared with that obtained with mupirocin treatment [182]. Honey has also been widely explored as a tissue-regenerative agent. In this case, it has been applied directly to the wound or in combination with traditional wound dressings, which allow honey to remain in direct contact with the wound bed, thus providing a persistent and long-term release of antimicrobial agents to contribute to all stages of wound healing. Furthermore, the presence of reactive oxygen species (ROS) such as H_2_O_2_ has been shown to promote wound healing by promoting cellular repair processes and tissue regeneration [10,183]. Anyway, some limitations exist, such as being absorbed by the dressing, poor penetration into the wound site, and short-term antimicrobial action. To address these issues, tissue engineering approaches have been developed, such as its formulation in electrospun fibers and hydrogels [184,185,186,187,188]. Collectively, the use of honey, honey-derived, and honey-inspired products in tissue engineering applications, combined with other biomaterials, may enable its use in a variety of other clinical situations outside wound care, where the combination of antimicrobial properties and tissue regeneration is desirable.

##### The Case of Surgihoney Reactive Oxygen (SHRO)

The first therapeutic agent based on the oxidative activity of ROS was a pharmaceutical honey gel for treating wounds, referred to as Surgihoney Reactive Oxygen (SHRO). Particularly, SHRO is an engineered, sterilized honey created to act as a preventive antimicrobial agent for soft tissue infections. SHRO, deriving from natural organic honey from different origins, is capable of providing a constant level of ROS over a prolonged period of time when topically applied to a wound. Subsequently, ROS induce OS due to •OH production and inhibit the essential metabolic procedure for bacterial growth [189]. As discussed in more in the subsequent sections, the antimicrobial activities of SHRO are mainly due to the generation of H_2_O_2_ [190]. Other formulated honey prototypes (PT1 and PT2) were designed to further increase the generation of H_2_O_2_. Subsequently, honey ROS-based antibiotic agents differently formulated, such as sprays, nebulizers, and infusions, that employ this mechanism are being developed and may be particularly useful for delivering ROS to other clinical sites. Nowadays, there are many types of therapeutic honey on the market (e.g., Paterson’s curse, Rosemary, Manuka, Thyme, Revamil, Rewa Rewa, heather honey, Khadi, Kraft honey, Multifloral, and Medihoney) [191,192]. Table 11 summarizes a clinical register using SHRO in various complex and severe infections [4].

With the rising age of the population in many countries, and the global epidemic of obesity and type 2 diabetes [4], the disease of chronic soft tissue lesions is becoming enormous. Most chronic breaks in the skin often become colonized with bacteria [193], which regardless of their pathogenicity, play an essential role in slowing tissue healing, establishing biofilm, and resulting in wound slough and an unpleasant odor [193]. As reported in Table 11, in these cases where often available antibiotics are no longer functioning, the effects of SHRO on bioburden and biofilm [189] can be of great help. In fact, the early use of ROS in such lesions can control bioburden and biofilm, thus sparing conventional antibiotic use and supporting infection control [189,194,195]. In clinical studies, ROS therapy via SHRO has demonstrated satisfactory safety and tolerability and is clinically and cost-effective in practice [194,196]. Most outstandingly, SHRO has demonstrated impressive capacity to clean up bacterial bioburden and biofilm in chronic wounds, being also active on MDR bacteria such as *P. aeruginosa*, MRSA, and vancomycin-resistant enterococci (VRE) present in an ischemic ulcer [4].

##### Medical Grade Honey for Clinical Applications

Medical-grade honey (MGH) intended for clinical application must be sterilized by gamma irradiation to destroy potentially present bacterial spores, including those of *Bacillus* spp. and *Clostridium botulinum*, which could cause wound botulism or gangrene [197]. Several types of honey, including MGH, have recently been re-introduced into modern medicine. There is no clear definition of MGH, but according to a study by Hermann et al., MGH should satisfy the criteria reported in Figure 4 [198].

Manuka and Revamil^®^ are the major medical-grade honeys currently approved for clinical application. Manuka honey is produced from the manuka bush (*Leptospermum scoparium*), a native of New Zealand and Australia. The raw honey used as a source for Revamil^®^ is instead produced by a standardized process in greenhouses. Since Revamil^®^ honey is registered as a medical device for applications in wound healing and not as an antimicrobial agent, the antimicrobial activity is not specified for individual batches of this honey. However, in a quantitative liquid bactericidal assay, both Revamil^®^ and manuka honeys demonstrated potent bactericidal activity [199,200], with manuka honey being the most performant against *S. aureus*, *B. subtilis*, and *P. aeruginosa*. On the contrary, these honeys had identical bactericidal activity against *E. coli*. The following Table 12 reports the approved and already commercially available wound healing products based on honey.

#### 3.3.2. Antibacterial Mechanisms of Honey

The antimicrobial activity of the majority of honeys is mainly due to its capability to generate high levels of hydrogen peroxide (H_2_O_2_) [10,21,190,223,224,225,226] by the oxidative action of the enzyme glucose oxidase (GOx) on glucose. Secreted into the nectar by bees during the preparation of honey, GOx oxidizes glucose to gluconic acid, thus producing H_2_O_2_ [10,226,227,228,229,230]. The enzyme presents no activity in raw honey due to a lack of free water. Therefore, to initiate the peroxide-dependent antimicrobial mechanism, the honey needs to be diluted. Other important antimicrobial features responsible for the non-peroxide activity of honey include low water content (osmotic effect), low pH (acidic environment), phenolic compounds, bee defensin-1 (Def-1), and methylglyoxal (MGO) (in *Leptospermum*-derived manuka honey). Table 13 summarizes the factors that confer honey its antibacterial effects.

Briefly, sucrose captured by bees from flowers is broken down via diastase and invertase enzymes into glucose and fructose. Glucose oxidase (Gluox), secreted by the bee’s hypopharyngeal glands, in the presence of O_2_ and sufficient H_2_O, oxidizes glucose, forming gluconolactone/gluconic acid, which make honey acidic and H_2_O_2_ [234]. H_2_O_2_ is the most responsible for honey’s antimicrobial activity, killing pathogens through DNA damage and being destructive to several cellular targets [234]. Interesting, the antimicrobial effect of hydrogen peroxide in honey increases upon dilution, enabling the glucose oxidase enzyme to bind to glucose more readily, thus resulting in a continuous production of hydrogen peroxide [226]. Moreover, honey, predominantly due to gluconolactone/gluconic acid formation, is acidic with an average pH of 3.91 (ranges between 3.4 and 6.1), which makes it powerful against microbial strains, preventing their growth. Bee Def-1 is an antibacterial peptide originating in the bee’s hypopharyngeal gland and identified in bee hemolymph (the bee blood system) [235]. Within the bee, it acts as an innate immune response, exhibiting activity against fungi, yeast, protozoa, and both Gram-positive and Gram-negative bacteria [200]. Importantly, bee Def-1 is mainly effective against Gram-positive bacteria, most notably *B. subtilis*, *S. aureus*, and *Paenibacillus larvae*, while it has limited effectiveness against MDR organisms [179]. Although the full mechanism of action for bee Def-1 has not been elucidated, defensin proteins from other species have been shown to create pores within the bacterial cell membrane, resulting in cell death [236]. Bee Def-1 demonstrated to play an important role in wound healing, through stimulation of MMP-9 secretions from keratinocytes [237]. It interferes with bacterial adhesion to surfaces, or in the early biofilm stage, by inhibiting the growth of attached cells and by altering the production of extracellular polymeric substances (EPS). MGO is generated in honey during storage by the non-enzymatic conversion of dihydroxyacetone, a saccharide found in high concentrations in the nectar of *Leptospermum* flowers [231]. The antimicrobial activity of MGO is attributed to alterations in bacterial fimbriae and flagella, which prohibit the bacterium’s adherence and motility. Honey is a super-saturated solution of sugars. The strong interaction between these sugars and water molecules prevents the abundance of free water molecules (low water activity) available for microbes to grow [232]. Finally, the combination of different phenols acts as an enhancer of honey’s antimicrobial efficacy. Produced as plant secondary metabolites, these bioactive compounds are transferred from the plant to the honey by bees and have been identified as the major reason for the health-promoting properties of honey [233]. In alkaline conditions (pH 7.0–8.0), polyphenols can display pro-oxidative properties, inhibiting microbial growth by accelerating hydroxyl radical formation and oxidative strand breakage in DNA. They could also support the production of considerable amounts of H_2_O_2_ via a non-enzymatic pathway.

### 3.4. ROS Formation Induced by Antibacterial Hyperbaric Oxygen Therapy

Differently from antibacterial honey therapy, which is a topical treatment, antibacterial hyperbaric oxygen therapy (HBOT), which is part of hyperbaric medicine, is a systemic method to treat soft tissue infections [57]. Particularly, in a typical HBOT treatment, the patient (mono-place) or more than one patient (multi-place) inhale 100% O_2_ for a specified time in a pressurized chamber [238]. From the lung, the inhaled oxygen is delivered to the whole body, where it induces the formation of ROS, which overwhelm the antioxidant defenses of the facultative anaerobic bacteria and aerobic bacteria, causing lipid peroxidation and membrane disruption, DNA injury, protein dysfunction, and death. A pressure greater than 1.4 atmosphere absolute (ATA) is necessary to have an effective antibacterial effect against some facultative anaerobic bacteria and aerobic bacteria [238]. In addition to being directly bactericidal via ROS formation, it has been reported that HBOT enhances the antimicrobial effects of the immune system, such as those of leukocytes in hypoxic wounds (Figure 5) [239].

Moreover, HBOT potentiates the antibacterial effects of some antibiotics, such as imipenem and tobramycin (Figure 6).

Hyperoxia (98% O_2_ at 2.8 absolute ATA—approximately 284.6 kPa) has been shown to enhance the effect of nitrofurantoin, sulfamethoxazole, trimethoprim, gentamicin, and tobramycin in *E. coli* strains (serotype 018 and ATCC 25922) [241]. Two works by Kolpen et al. report that the combination therapy of ciprofloxacin and HBOT may be potentially beneficial for the eradication of infections caused by the biofilm-forming *P. aeruginosa* [242,243]. Anyway, HBOT did not show any additive or synergistic effect with other antimicrobial agents, such as distamycin and rifampicin [57]. Very recently, it has been reported that wild strains of *Pseudomonads*, *Burkholderias*, and *Stenotrophomonads* stopped growing under hyperbaric conditions at a pressure of 2.8 ATA of 100% oxygen [241]. HBOT is used as a primary or alternative method for the treatment of infections such as diabetic foot infections, surgical site infections, gas gangrenes, osteomyelitis, and necrotizing compartments [240]. In addition to the bactericidal effects, HBOT suppresses the production of clostridia alpha toxin in gas gangrene diseases. HBOT has also demonstrated anti-inflammatory effects that may play a significant role in decreasing tissue damage and infection expansion. While generally safe, HBOT may have side effects such as ear discomfort, sinus pain, and temporary nearsightedness. Other contraindications include untreated pneumothorax (collapsed lung), certain medications, and claustrophobia. Although patients treated by HBOT need careful pre-examination and monitoring, when safety standards are strictly tracked, HBOT can be considered a suitable procedure to treat severe infections sustained by MDR bacteria with an acceptable rate of complication.

#### Clinical Application of HBOT in Infections

Several studies have evidenced that HBOT, either alone or as an accessory treatment, can be a valuable therapeutic option to cure patients with several diseases, including difficult-to-treat infections (Table 14).

Basically, HBOT strongly improves the levels of O_2_ concentration in blood and the oxygen pressure both in blood and tissues (2000 mmHg and 500 mmHg, respectively), determining hyperoxia conditions, which provide beneficial effects in patients suffering from several diseases as reported in Table 14. HBOT was used to adjust immunology and maintain the durability of an allograft [264]. HBOT has demonstrated beneficial effects on the vascular endothelium, thus promoting angiogenesis and induced partial high tensions of O_2_ in circulating plasma, thus stimulating O_2_ dependent collagen matrix formation, which is an essential phase in wound healing [265]. On the other hand, HBOT effects on sepsis, urinary tract infections, and meningitis are not well known so far. Unequivocally, the most frequent clinical application of HBOT remains for several skin soft tissue infections and osteomyelitis infections, which are associated with hypoxia, caused by anaerobic infections due to antibiotic resistant bacteria [266,267]. Currently, HBOT is considered both alone and in combination with different antibacterial treatments as a relevant option for solving several cute or chronic diseases [260,262,268,269]. Although animal studies have shown the inhibitory effect of HBOT on inflammation and apoptosis after cerebral ischemia [270,271], clinical trials on humans have not shown any significant benefit. Anyway, it has been indicated that HBOT can improve some neuropsychological and inflammatory outcomes, especially in stroke patients, within the first few hours [272,273]. Furthermore, studies on animals have shown that HBOT is associated with reduced blood-brain barrier breakdown, reduced cerebral edema, improved cerebral oxygenation, decreased intracranial pressure, reduced oxidative burden, reduced metabolic derangement, and increased neural regeneration [271,272,274]. The following Table 15 contains an overview of some clinical studies investigating the application of HBOT for different infections, while Table 16 collects the most relevant studies concerning, specifically, the clinical application of HBOT in surgical site infections (SSIs) categorized based on the type of SSI or surgery. Sternal wound infections following cardiac surgery, SSIs following neuromodulation or neuro-muscular surgery, and SSIs following male-to-female gender affirmation surgery (urogenital surgery) have been included.

## 4. Possible Novel Methods to Induce ROS Formation: Our Proposal

### 4.1. Environmental Persistent Free Radicals (EPFRs)

Environmentally persistent free radicals (EPFRs) are defined as long-living organic free radicals stabilized on or inside particles [305]. EPFRs are persistent because of the protection provided by the particles containing them and the presence of transition metals, thus having lifetimes that are exceptionally longer (from days to years) than other free radicals. These surface-bound radicals are found in contaminated soil, tar balls, and cigarette smoke and primarily form during thermal processes such as pyrolysis and combustion of organic materials, waste incineration, and photoactivation [306]. In fact, the byproducts from these processes, such as phenols and aromatic polycyclic hydrocarbons, provide a breeding ground for EPFR generation. In fact, EPFRs are also found in the matrix of ultrafine and airborne fine particulate matter (PM), which is emitted into the environment by both natural and anthropogenic processes, including coal combustion emissions (16.8%), vehicular emissions (32.1%), industrial processes (11.7%), dust storms (27.2%), and nitrates (3.4%). PM is a mixture of organic species, inorganic species, solid, and liquid components of metals that have the tendency to form radicals with or without sunlight photoactivation [306,307]. The environmental factors affecting the EPFRs’ formation, lifetime, and abundance include precursors, temperature, light irradiation, the presence of metals, temperature, pH, humidity, thermal processing time, and oxygen [308]. EPFRs are categorized as no decay, low decay, and fast decay radicals. The no-decay EFFRs are unpaired electrons delocalized over aromatic bonds entrapped inside PM. Phenoxyl radicals are fast-decay EPFRs, while semiquinone radicals are subjected to slow decay [309]. EPFRs can produce reactive oxygen species, including hydroxyl radicals, which induce oxidative stress in living organisms, posing adverse environmental and human health effects. The atmospheric oxygen is the only sink for stable free radicals, converting them into particles and/or metal stabilized molecular species, thus decaying [306]. EPFR decay in the atmosphere depends on the reaction of EPFRs with molecular oxygen. By reacting with atmospheric oxygen, they generate high levels of reactive oxygen species (ROS) via electron transfer, such as hydroxyl radicals and superoxide anion radicals, thereby inducing cellular oxidative stress [23,24]. For this reason and their possible redox recycling, EPFRs are emerging as environmental pollutants with a ROS-dependent significant toxicity to organisms, including humans, plants, animals, and microorganisms. Anyway, recent research has also explored their potential for degrading organic environmental contaminants. By activating hydrogen peroxide or persulfate, EPFRs produce ROS species such as superoxide, singlet oxygen, or OH^−^, thus inducing the degradation of environmental organic pollutants [306] Collectively, although EPFRs are long-lived environmental pollutants harmful to the environment and living beings, their capacity to activate ROS generation, if properly controlled, can make them a wide range of tools for environmental remediation.

### 4.2. Biochar-Derived Persistent Free Radicals (PFRs)

Biochar (BC) is a carbonaceous material obtained by the pyrolysis of different vegetable and animal biomass feedstocks at 200–1000 °C in the limited presence or absence of O_2_. BC has demonstrated a broad prospective use in the treatment of environmental pollutants and in soil amendment. It has been used in photocatalytic and photothermal systems for photothermal conversion, to construct electrical and thermal devices, as well as 3D solar vapor-generation devices for water desalination [24]. All these potentials are due to its high surface area and rich pore structure, which determine its great physical absorptivity [23]. Additionally, they also depend on the chemical characteristics of BC, which in turn depend on the type of biomass used to produce BC, the original biomass chemical composition, and pyrolysis conditions [310,311]. Whereas, starting in 2014 (Figure 7), the presence of persistent free radicals (PFRs) in BC deriving from lignocellulosic biomasses, like the radicals previously detected in combustion-generated particulate matter (PMs), sediments, and contaminated soils, known as environmental persistent free radicals (EPFRs), has been reported.

As EPFRs, such reactive species can remain stable for months or years and play a crucial role in the capacity of BC to degrade different types of xenobiotics and pollutants by oxidative reactions via ROS formation. Unlike other free radicals, including ROS, PFRs are resonance-stabilized since they are bound to the external or internal surface of solid particles of BC [24]. If BC is conserved under vacuum, the lifetime of PFRs could be infinite (no decay radicals), while when air-exposed, they react with molecular oxygen in the air and decay over time, thus producing ROS. Similarly, in aqueous systems, PFRs act as transition metals such as Fe^2+^, forming ROS as well [312,313,314,315]. PFRs are categorized into three classes, i.e., oxygen-centered PFRs (OCPFRs), carbon-centered PFRs (CCPFRs), and oxygenated carbon-centered radicals (CCPFRs-O). The possible presence of PFRs on a BC, their type, and their concentrations are significantly affected by pyrolysis conditions, biomass types, the elemental composition of pristine biomass, and the presence of external transition metals, as detailed in Table 17.

#### 4.2.1. Proposed Mechanisms for PFR Formation during Biomass Pyrolysis

The actual mechanism by which PFRs form during pyrolysis remains has not been fully clarified. However, transition metals capable of electron transfer and substituted aromatic molecules present in the lignin component of pristine biomass have been recognized to be essential for PFR formation. Anyway, high concentrations of PFRs have also been detected in products obtained by the pyrolysis of non-aromatic cellulose in the absence of transition metals [319]. Collectively, PFRs can form by different pathways, including or without the presence of transition metals, and once formed, PFRs could be either only surface-stabilized or surface-stabilized in metal-radical complexes [321]. Figure 1 (concerning lignin) and Figure 2 (concerning cellulose and emicellulose) report the possible chemical paths by which PFRs may form.

Since it is out of scope of this paper, a detailed discussion on the mechanisms reported in Figure 1 and Figure 2 has been avoided. Readers particularly interested can find major information in a very recent review [24].

#### 4.2.2. Possible Activities of PFRs and Our Proposal

PFRs formed in BC during combustion of lignocellulosic biomasses, either in the presence or absence of external transition metals, could promote several beneficial reactions, such as PFR-mediated remediation and degradation of organic and inorganic pollutants by different actions and mechanisms, including oxidative and reductive processes. PFRs on BC can activate hydrogen peroxide (H_2_O_2_) or oxygen (O_2_), as well as persulfate (S_2_O_8_^2−^), to produce different radical and not radical oxygenated species (ROS) capable of efficiently degrading organic contaminants by oxidative mechanisms, as ROS generated by the previously reported antimicrobial therapies are bactericidal to pathogens inducing OS via ROS stimulation. Therefore, we thought that ROS induction using BC-derived PFRs, whose type, concentration, and reactivity can be tunable under pyrolysis conditions, could be a novel method to form ROS for a possibly more selective BC-based antibacterial oxidative therapy. In this regard, in a recent review on BC-derived PFRs, a random selection of the main experimental works regarding the applications of PFRs found in BCs conveyed in the last five years (2019–2023) has been reported. Among the reported PFR applications, three regarded their use as antibacterial agents (Table 18), thus supporting our idea.

BC employed was derived from the pyrolysis of sludge, the *Caragana korshinskii* plant, and pinewood. In these processes, the electron transfer promoted by PFRs of diverse nature generated ROS such as SO_4_^•−^, •OH, •O_2_^−^, and •O_2_H, which carried out the oxidative degradation of different organic pollutants, including drugs, dyes, antibiotics, and hormones, and showed antibacterial effects against *E. coli* and *S. aureus*.

## 5. Conclusions

Upon the colonization of the host cell during infection, the maintenance of redox homeostasis (RH) is a key process for bacterial survival and for escaping the oxidative stress physiologically generated by macrophages to oppose the development of the infection. Pathogens succeed in strictly controlling RH through a mechanism based on different redoxins and low-molecular-weight-thiol molecules. Therapeutic strategies based on the capacity of different compounds or methods to cause ROS and RNS hyper-generation during phagocytosis to unbalance bacterial redox defenses and stop host cell colonization have a great potential to solve the increasing problem of antibiotic-resistant infections. ROS have demonstrated to be effective in inhibiting clinically important microbial pathogens by lipid peroxidation, thus damaging their membranes, harming DNA, and impairing protein functions. It has been observed that certain traditional antibiotics and alternative antimicrobials, including nanomaterials, as well as their combination, induce ROS as a secondary or main mechanism of antibacterial effect. Additionally, in the worrying scenario of an increasing global emergence of difficult-to-treat infections due to bacterial resistance to available antibiotics, old procedures that inhibit microbial growth by forming ROS, such as HBOT and medical honey, have been reinvigorated. Together with the more recent PDT, they are, in fact, currently successfully employed for the treatment or prevention of soft tissue infections and chronic ulcerations. The development of resistance to these methods has not been reported, but unfortunately, since ROS-mediated OS is destructive to eukaryotes as well their clinical application to treat systemic infections is at present impracticable. Further studies aimed at identifying novel delivery techniques for using ROS with superior selectivity for microbial pathogens are required. As our contribution to this challenge, we have now proposed BC-associated PFRs as a promising novel, low-cost, and eco-friendly method for ROS generation to be studied for the oxidative inhibition of MDR pathogens and as a potential treatment for a wide range of infections. Greater knowledge concerning the proper pyrolysis conditions employed to obtain the type of PFRs more suitable for this purpose and in optimized concentration could make this ROS-delivering method more selective for bacterial pathogens.

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
