# Peer review of "Reactive Oxygen Species (ROS)-Mediated Antibacterial Oxidative Therapies: Available Methods to Generate ROS and a Novel Option Proposal"

_ijms, 2024, doi:10.3390/ijms25137182_

Round 1
Reviewer 1 Report
Comments and Suggestions for Authors
This manuscript reports the physiological and pathological production and regulation of ROS, and its mechanism for antimicrobial functions. The future possible applications of BC-related PFRs have also been discussed. This review covers a broad range of ROS production and its possible role for antibacterial applications, which contains a lot of summaries in the related field. Some issues still need to be solved before publication.
Specific comments:
1. In Figure 1, please describe the figure content in the main text to make what happens clear.
2. It would be better to show the advantages and disadvantages of common photosensitizers used in clinical trials.
3. Schemes with chemical structures have low resolution, please improve them.
Author Response
Reviewer 1
This manuscript reports the physiological and pathological production and regulation of ROS, and its mechanism for antimicrobial functions. The future possible applications of BC-related PFRs have also been discussed. This review covers a broad range of ROS production and its possible role for antibacterial applications, which contains a lot of summaries in the related field. Some issues still need to be solved before publication.
Specific comments:
- In Figure 1, please describe the figure content in the main text to make what happens clear.
We thank the Reviewer for this comment. As asked, a more detailed explanation of the events described in Figure 1 has been included in the text. Please, see lines 344-349.
- It would be better to show the advantages and disadvantages of common photosensitizers used in clinical trials.
We thank the Reviewer for this comment. As asked, the advantages and disadvantages of the most common PSs used in clinical treatments and/or trials have been included in the main text with the proper references. Please, see at lines 536-563 and Refs. 98-104.
- Schemes with chemical structures have low resolution, please improve them.
The schemes (Scheme 1 and Scheme 2) are those already published on MDPI journals and the resolution of the chemical structures included, have been already approved as suitable for publication.
Generally, we thank a lot the Reviewer for his/her positive comments.

Reviewer 2 Report
Comments and Suggestions for Authors
Reviewer’s comments
Manuscript ID: ijms-3048374
Type of manuscript: Review
Title: Reactive Oxygen Species (ROS)-Mediated Antibacterial Oxidative Therapies: Available Methods to Generate ROS and a Novel Option Proposal
The research work entitled “Reactive Oxygen Species (ROS)-Mediated Antibacterial Oxidative Therapies: Available Methods to Generate ROS and a Novel Option Proposal” presents on the potential of exploiting ROS-inducing strategies in biomedical treatments, particularly bacterial diseases. Additionally, the study revealed detailed biochemical mechanisms and identified potential materials or natural substances that induce ROS. This work has properly organized the current academic literature on ROS induction and biomedical applications. However, some concepts, the logic of article paragraphs, and quotation sentences for tables proposed by the author in the article is poor in presentation. I still believe that this work can make a great contribution for researchers to report on the recent development and application potential of emerging materials with ROS-inducing ability. I suggested this work may be accepted for publication in “International Journal of Molecular Sciences” after a “major revision”. Specific comments and general comments are given below:
Specific comments
1. The Introduction section is generally well written, but the second half can be conceptually confusing. To enhance clarity, it is suggested to divide it into sections as follows: Lines 67-84, Lines 85-94, Lines 95-111, Lines 112-120, and Lines 120-146.
2. Fig. 1, Fig.7, Scheme 1, and Scheme 2, to modify self-published figures, it is strongly recommended to obtain permission from the journal and ensure that the proofs are cited in accordance with the publisher’s policy.
3. Lines 151-152 and Lines 158-160 are somewhat incoherent. It is recommended to rewrite these lines and then integrate them into the paragraphs from Lines 166-175. Additionally, it would be beneficial to re-segment Lines 176-192 and Lines 193-224. A similar issue is present in Section 3.2.4 (Line 547) and should be addressed as well.
4. I believe the information arrangement in Table 4 could be more reader-friendly, and I suggest reorganizing and displaying the information more clearly. For instance, the Effect field should not be left blank for the enzymatic process. Additionally, there should not be a blank line in the middle of the sentences in Lines 230-232.
5. Regarding the arrangement of section 3.1, the single sentence provided may not fully convey the intended meaning. It might be helpful to expand or clarify this sentence for better understanding.
6. For section 3.1.2, it might be beneficial to include an additional paragraph to explain the induction of ROS by emerging materials such as carbon dots (CDs), particularly those derived from processed food or edible substances. Additionally, it is suggested to consider removing the titles in Lines 365, 374, 392, and 401.
7. Table 12 contains some disease fields that are blank and formatted incorrectly. Additionally, there are symbols such as ° and * that lack descriptive meaning. It would be beneficial to revise these aspects for clarity and accuracy.
General comments
- Line 884, please remove Ref.
- Table 15, Please explain the meaning of the symbols, ≠, ⇑, ⇓, and it is recommended that all table symbols be consistent.
Author Response
Manuscript ID: ijms-3048374
Type of manuscript: Review
Title: Reactive Oxygen Species (ROS)-Mediated Antibacterial Oxidative Therapies: Available Methods to Generate ROS and a Novel Option Proposal
The research work entitled “Reactive Oxygen Species (ROS)-Mediated Antibacterial Oxidative Therapies: Available Methods to Generate ROS and a Novel Option Proposal” presents on the potential of exploiting ROS-inducing strategies in biomedical treatments, particularly bacterial diseases. Additionally, the study revealed detailed biochemical mechanisms and identified potential materials or natural substances that induce ROS. This work has properly organized the current academic literature on ROS induction and biomedical applications. However, some concepts, the logic of article paragraphs, and quotation sentences for tables proposed by the author in the article is poor in presentation. I still believe that this work can make a great contribution for researchers to report on the recent development and application potential of emerging materials with ROS-inducing ability. I suggested this work may be accepted for publication in “International Journal of Molecular Sciences” after a “major revision”. Specific comments and general comments are given below:
Specific comments
- The Introduction section is generally well written, but the second half can be conceptually confusing. To enhance clarity, it is suggested to divide it into sections as follows: Lines 67-84, Lines 85-94, Lines 95-111, Lines 112-120, and Lines 120-146.
We thank the Reviewer for his/her suggestions. As asked, the Introduction section has been improved by dividing lines 67-146 in subsections as proposed by the Reviewer. The tiles of new subsections can be found at lines 67, 86, 95, 100 and 118.
- Fig. 1, Fig.7, Scheme 1, and Scheme 2, to modify self-published figures, it is strongly recommended to obtain permission from the journal and ensure that the proofs are cited in accordance with the publisher’s policy.
As underlined by the Reviewer, Figures 1 and 7 contain images previously published by us on Antioxidants (Figure 1) and Toxics (Figure 7), which are MDPI journals, whose published articles are Open-Access articles distributed under the terms and conditions of the Creative Commons Attribution License (CC BY). No permission is necessary neither from the authors or from the journal for their reproduction. The same is for Scheme 1 and Scheme 2 already published on Toxics. The proper citations have been included in the manuscript.
- Lines 151-152 and Lines 158-160 are somewhat incoherent. It is recommended to rewrite these lines and then integrate them into the paragraphs from Lines 166-175. Additionally, it would be beneficial to re-segment Lines 176-192 and Lines 193-224. A similar issue is present in Section 3.2.4 (Line 547) and should be addressed as well.
We thank a lot the Reviewer for his/her suggestions which have been fully addressed. The revision work can be appreciated in lines 156-158,164-171, 182-208, 209-213, 218-224, 237-242 etc. Section 3.2.4. and related Table have been fully reorganized.
- I believe the information arrangement in Table 4 could be more reader-friendly, and I suggest reorganizing and displaying the information more clearly. For instance, the Effect field should not be left blank for the enzymatic process. Additionally, there should not be a blank line in the middle of the sentences in Lines 230-232.
We thank a lot the Reviewer for his/her suggestions which have been fully addressed. Table 4 has been completely reorganized in the new Tables 4a and 4b. The issue with lines 230-232 (original version) has been addressed.
- Regarding the arrangement of section 3.1, the single sentence provided may not fully convey the intended meaning. It might be helpful to expand or clarify this sentence for better understanding.
As asked the sentence has been remodulated for better clarity. Please, see lines 321-325.
- For section 3.1.2, it might be beneficial to include an additional paragraph to explain the induction of ROS by emerging materials such as carbon dots (CDs), particularly those derived from processed food or edible substances. Additionally, it is suggested to consider removing the titles in Lines 365, 374, 392, and 401.
The requested paragraph on CDs has been added with related additional references (lines 393-396 and 423-443, Refs. 79-84) and the signalled titles have been removed (lines 397, 406, 444, and 453).
- Table 12 contains some disease fields that are blank and formatted incorrectly. Additionally, there are symbols such as ° and * that lack descriptive meaning. It would be beneficial to revise these aspects for clarity and accuracy.
The issues signalled for Table 12 have been addressed, and the descriptive meaning of ° and * symbols has been added in the Table title (lines 896-898).
General comments
Line 884, please remove Ref.
Done.
Table 15, Please explain the meaning of the symbols, ≠, ⇑, ⇓, and it is recommended that all table symbols be consistent.
We thank a lot the Reviewer for his/her comment. Accordingly, we have inserted the due explanation in the foot notes of Table 15 (lines 1016-1017).

Round 2
Reviewer 2 Report
Comments and Suggestions for Authors
I am satisfied with the revised version.